# Cervical Cancer Screening Recommendations: Now and for the Future

**DOI:** 10.3390/healthcare11162273

**Published:** 2023-08-11

**Authors:** Marissa Rayner, Annalyn Welp, Mark H. Stoler, Leigh A. Cantrell

**Affiliations:** Department of Obstetrics and Gynecology, University of Virginia, Charlottesville, VA 22908, USA

**Keywords:** cervical cancer screening

## Abstract

Cervical cancer is the fourth most common cancer worldwide, with over 600,000 new cases annually and approximately 350,000 cancer-related deaths per year. The disease burden is disproportionately distributed, with cancer-related mortality ranging from 5.2 deaths per 100,000 individuals in highly-developed countries, to 12.4 deaths per 100,000 in less-developed countries. This article is a review of the current screening recommendations and potential future recommendations.

## 1. Introduction

Cervical cancer is the fourth most common cancer worldwide, with over 600,000 new cases annually and approximately 350,000 cancer-related deaths per year [1]. The disease burden is disproportionately distributed, with cancer-related mortality ranging from 5.2 deaths per 100,000 individuals in highly-developed countries, to 12.4 deaths per 100,000 in less-developed countries [1].

### 1.1. Epidemiology

Even within developed countries such as the United States (US), there continue to be stark population-level differences in the incidence and mortality of cervical cancer. Hispanic and Black populations in the US are at significant socioeconomic disadvantage compared to the White population [2], leading to obstacles in healthcare access and the ability to obtain appropriate screening and preventative services; this is consequently mirrored in US cervical cancer statistics. In the US, Hispanic women have the highest rates of cervical cancer incidence, at 10 women per 100,000, followed closely by American Indian, Alaskan Native, and Black women [3]. Black women have the highest mortality related to cervical cancer in the US, at 3 per 100,000 as compared to 2 per 100,000 in white, non-Hispanic patients [3]. Disproportional disease burdens are reflected on both international and national levels [1,4]. The differences in distribution are considered multifactorial, including differing prevalence of risk factors, levels of disease awareness, access to screening, treatment availability, and vaccination programs. 

### 1.2. Cervical Cancer, Human Papillomavirus (HPV), and Vaccination

Human Papillomavirus (HPV), a sexually-transmitted viral infection, has a well-established causative relationship to cervical cancer. HPV is the leading cause of cancers of the cervix, vulva, vagina, penis, anus, and oropharynx. Risk factors for cervical cancer are similar to risks related to exposure and persistence of HPV. These risk factors include: multiple sexual partners, early age in initiation of sexual intercourse, high parity, low socioeconomic level, and tobacco use [5,6]. Several risk factors are linked to an increased probability of HPV exposure, such as multiple sexual partners and sexual intercourse at a young age. Others, such as tobacco use or HIV, have been suggested given their role in immune suppression possibly prompting cervical carcinogenesis [7,8]. It is postulated that suppressed immune function results in a decreased ability to clear HPV, leading to persistent infection and increased risk of cervical dysplasia (cervical intraepithelial neoplasia (CIN)) lesions [9]. Thus, the unique relationship between HPV exposure, prolonged infection, and diminished immune response sets the stage for cervical carcinogenesis.

HPV itself is a widely prevalent group of viruses that has been estimated to cause over 95% of cervical cancers [10,11]. In 2018, one global meta-analysis found an HPV prevalence of 11.7%, with a bimodal age-specific distribution having peaks at <25 years and >/= 45 years of age [12]. Lifetime risk of infection is about 80% in sexually active women, however, a healthy person’s robust immune response is typically responsible for resolution of HPV infectivity before persistent cervical dysplasia or neoplasia develops [9]. 

Over 200 different variants of HPV have been described, some of which include high-risk HPV (HR-HPV) strains that are most strongly associated with oropharyngeal and anogenital cancers, with others considered low-risk HPV (LR-HPV) subtypes responsible for cutaneous and anogenital warts [13]. Specific HR-HPV subtypes are associated with up to 95% of all squamous cell cervical cancers, and one particular study found subtypes 16, 18, 31, 33, 35, 39, 45, 51, 52, 56, 58, 59, 68, 73, and 82 to be the majority of those involved [14]. In this same study, 50.5% of all cervical cancers positive for HPV were positive for HPV-16 and 13.1% positive for HPV-18, indicating that while many HR-HPV subtypes do exist, HPV-16 and 18 account for a majority of cervical cancers [14]. LR-HPV subtypes include HPV-6, 11, 40, 42, 43, and 44 among many others [14]. In 2018, the five most common subtypes of HPV worldwide were found to be HPV-16, 18, 31, 52, and 59—all of which have been found to be associated with cervical cancer and precursor lesions [12,15]. 

Understanding of the relationship between different HPV variants and carcinogenesis led to the development of targeted vaccines against the HR-HPV strains. The first prophylactic HPV vaccine was released in 2006 under the name Gardasil^®^, which initially protected against four HPV strains (6, 11, 16, and 18) [16]. Since then, several new vaccines have been released with six currently licensed HPV vaccines utilized internationally and recommended by the World Health Organization (WHO) for use. These include three bivalent vaccines (Cervarix^®^ (GlaxoSmithKline Biologicals, Rixensart, Belgium), Cecolin^®^ (Xiamen Innovax Biotech Co. Ltd., Xiamen, Fuijan Providence, China), and Walrinvax™ (Walvax Biotechnology Co., Kunming, Yunnan Providence, China)), two quadrivalent vaccines (Gardasil^®^ (Merck & Co, Rahway, NJ, USA) and Cervavac^®^ (Serum Institute of India, Pune, India)), and one nonavalent vaccine (Gardasil9^®^ (Merck & Co, Rahway, NJ, USA)). All offer protection against HR-HPV types 16 and 18, with additional protection against types 6 and 11 offered by quadrivalent vaccines. The nonavalent vaccine also offers protection against HR-HPV types 31, 33, 45, 52, and 58. All vaccines are indicated for use in females aged 9 years or older, up to 26 to 45 years of age; or are approved for use in male individuals depending on the vaccine [17]. With vaccine use, up to 70% of cervical cancers related to HPV can be prevented—this increases to as high as 96.7% in the nonavalent vaccine, as coverage of more HR-HPV strains confers even greater protection [18,19]. Unsurprisingly, vaccination has proven essential in the primary prevention of cervical cancer on a global scale.

### 1.3. Screening

Vaccination is the only way to primarily prevent cervical cancer, while cervical cancer screening measures are essential for secondary prevention. The first cervical cancer screening test was developed by George Papanicolaou and Herbert Traut, which they described in their book Diagnosis of Uterine Cancer by Vaginal Smear in 1943 [20]. Although many discoveries and developments have been made since the 1940s, the basis of the “Pap smear”, or analysis of cervical cytology remains integral to screening today. In the early 2000s, development of liquid-based cytology was introduced as an alternative and now preferred method of performing a Pap smear [21]. Several advantages led to its widespread use for screening, including cost-effectiveness, decreased frequency of unsatisfactory sampling, and ability to be combined with HR-HPV co-testing on the same sampling system [22,23]. Use of cytology has led to marked decreases in cervical cancer incidence due to the integration of escalating interventions such as colposcopy, loop electrosurgical excision procedure (LEEP), or cold knife conization when abnormal Pap smears result. While the sensitivity of a single Pap smear alone to correctly identify high-grade lesions (defined as CIN2/CIN3) has been estimated in the range of 50–55%, it is generally a well-tolerated, quick procedure with benefits that typically outweigh the risks involved, which has led to its overall success as a screening tool when combined with treatment [24,25]. Other modalities of cervical cancer screening historically used in resource-scarce settings or less-common settings include visual inspection with acetic acid (VIA) and visual inspection with Lugol’s iodine (VILI). VIA has been shown to have variable sensitivities, typically at the same level of or even higher than Pap smears [26,27,28,29]. However, its specificity is rather low, leading to an increased propensity for referrals for further workup than necessary [27]. There are less data surrounding VILI than VIA, but overall they both appear to be reasonable for screening in low resource regions where cervical cytology with Pap smear or HPV testing are unavailable [29].

The development and integration of HPV testing has led to a more robust screening modality for cervical cancer. Randomized controlled trials illustrate cervical HPV testing is more sensitive than cytology alone in the detection of precancerous lesions and provides a longer low-risk period for determination of intervals between screening tests [30,31,32,33]. Data also showed that HPV testing alone provided a 60–70% greater protection against cervical cancer than cytology alone due to its increased ability to accurately determine the risk of precancerous or cancerous lesions, leading to subsequent interventions and treatments [34]. Further studies have continued to affirm these findings, with the paradigm of cervical cancer screening shifting towards HPV testing as initial screening, and away from cytology alone, in recent years. 

There are several modalities by which the presence of HPV infection can be tested, including testing for the presence of DNA, RNA, proteins, or epigenetic biomarkers [35]. The most common tests utilized are DNA and mRNA testing, both of which have been proven to offer sensitivities above 0.90 which are appropriate for use in screening [36,37,38]. These tests involve endocervical or cervical sampling, typically performed by a trained health care provider [39]. Newer advancements in HPV testing have led to the development of self-collected sampling modalities that have been found to have similar rates of accuracy when compared to clinician-collected tests [40]. Self-sampling has the potential to increase reach to underserved populations, and new studies are emerging with promising evidence that this method is effective for access-limited populations when thoughtfully employed [41,42]. Self-sampling methods may also be a key tool for providers in the provision of trauma-informed care for patients with histories of sexual assault or the transgender population.

One of the newest approaches to determining the risk of cancer in an HPV-positive patient is dual-stain testing. Dual-stain testing measures the presence of p16 and Ki-67 proteins in the cervical cells sampled [43]. The presence of p16 is strongly linked to HPV infection, and Ki-67 is a biomarker for rapid cell division as seen in precancers and cancer [44]. Testing for the presence of these proteins has proven to be useful in triaging HPV-positive patients to aid in the determination of need for biopsy. An NCI study directly comparing standard cervical cytology to dual-stain testing for triage in HPV-positive patients found that dual-stain positivity was associated with a significantly higher risk of having at least a CIN2 lesion within 5 years compared to standard cervical cytology alone as a triage tool, as well as a significantly lower risk of having a CIN2 lesion or higher if dual-staining was negative compared to standard cytology [45]. Dual staining is not only a better predictor of the development of lesions in patients who test positive for HPV, but it also confers a 5-year interval if negative, which is more favorable than the 3-year interval of cervical cytology [45]. In March 2020, the US FDA approved the first dual-stain test for women who have tested positive for HPV [46]. An interesting area of development within this topic is the use of artificial intelligence (AI) for interpretation of dual-stain testing, which appears to have a comparable sensitivity and improved specificity in initial studies [47]. While the dual-staining testing modality is still in its early stages in terms of large-scale application, data is promising for its role in triaging HPV-positive patients and streamlining the process by which individuals are screened and evaluated for cervical cancer.

Today, guidelines for the prevention and screening for cervical dysplasia and malignancy have become convoluted and difficult for practitioners to comprehend. Below, we discuss the current state of cervical cancer screening guidelines, attempt to predict the changes that are likely in the future, and discuss barriers to such changes.

## 2. Discussion

### 2.1. Guidelines for Screening

Several different sets of cervical cancer screening guidelines are currently published and are compared in Table 1. On an international level, the WHO published an updated screening guideline in 2021 recommending cervical cancer screening in all women in the general population starting at age 30 with HPV DNA testing only rather than with cytology or VIA. The WHO recommends this screening be performed every five to ten years until the age of 50, where screening can be stopped after two negative tests at the recommended general population screening intervals. For individuals who are HIV positive, the WHO recommends screening with HPV DNA testing starting at age 25 every three to five years until age 50, when screening can be stopped after two negative tests at the HIV-specific screening intervals [48]. The European Commission released updated cervical cancer screening recommendations in November of 2022. Their recommendations included HPV testing for women aged 30 to 65 with an interval of five years or more, and consideration of utilizing self-sample kits in those who did not otherwise respond to screening invitations. They also suggested consideration of adapting the ages and intervals based on HPV vaccination status. These recommendations were made as part of an overarching set of guidelines aimed at increasing cancer screening in all qualifying individuals in European countries by the year 2025 [49,50]. The International Federation of Gynecology and Obstetrics (FIGO) recommends a more individualized approach to cervical cancer screening on a national or regional basis, taking into consideration the access, barriers, and resources available to different populations [51]. 

While there are minor differences in screening ages and intervals of the cervical cancer screening guidelines recommended by these international organizations, there has been an overall collective shift to HPV-DNA as the screening test of choice over the past two decades [48,59]. Cervical cytology still remains popular in some countries such as the United States, likely due to existing laboratories and other established screening infrastructure [56,58,60]. As new laboratories and systems are built to support HPV testing, a shift to align with the most up-to-date screening recommendations internationally is expected.

Larger differences are seen when looking at national guidelines. Some countries have unified national screening guidelines, while others follow several guidelines offered by various organizations. Implementation varies greatly as well, with some countries possessing well-established nationwide population-based screening programs, while others leave screening up to provincial or territorial programs, or none at all [61]. For example, the United States has several different major organizations with unique sets of cervical cancer screening guidelines. These organizations include the United States Preventive Services Task Force (USPSTF), American College of Obstetricians and Gynecologists (ACOG), American Cancer Society (ACS), American Society of Clinical Oncology (ASCO), and the American Society for Colposcopy and Cervical Pathology (ASCCP). A summary of these organizations’ guidelines is also included in Table 1. Despite well-established evidence that HPV testing alone is a superior method of screening for cervical cancer, guidelines around the world continue to differ in screening method, time interval, and age, among other aspects. These differences may be in part due to access obstacles, the routine delay in updating recommendations, and population adherence [61].

Creating a single universal cervical cancer screening guideline is a difficult task, which may explain why so many different versions currently exist. There may be several unique barriers that a single community may have to adhering to recommendations [62]. When considering the entirety of the world’s population, the creation of universal guideline is clearly quite a complicated undertaking. 

### 2.2. Resources: A Barrier to Implementation

Several barriers have been identified that complicate the implementation of universal cervical cancer screening; one major barrier is access to resources [63]. In even the most developed nations, reaching an entire population is difficult due to unequal distribution of access to resources [64]. Significant infrastructure is necessary to support a screening method [65]. Most HPV testing requires approved testing kits consisting of sample collection materials, specimen storage medium, and a method to transport samples to the appropriate laboratory for processing [39]. Once at a laboratory, there are various ways in which the different types of HPV tests are run by lab specialists based on testing protocols—sometimes this involves genetic amplification with PCR, other times it involves antibody hybridization and luminescence, to briefly characterize just two of the ways in which HPV can be detected [39]. There is a growing number of different HPV tests approved for use, as well as non-approved HPV testing kits. Approval of new HPV tests typically requires standardized comparison and validation against validated options [36]. The production, availability, and use of non-approved HPV tests may theoretically reach a wider population but are not preferable due to potential poor clinical validity; providers should aim to offer and utilize approved HPV tests for screening. Each of the numerous internationally approved HPV tests requires the appropriate structural support to process samples on a large scale for a community [36,65]. A newer alternative to the commonly used HPV testing laboratory approach is the use of point-of-care HPV testing, with processing on a smaller-scale clinic basis rather than central laboratory structure. A point-of-care HPV system allows for diagnosis and immediate initiation of next-step diagnostic tests or treatment, which is valuable for populations in which loss to follow-up is common [66]. A novel study performed in Papua New Guinea published in 2022 showed that development of a comprehensive point-of-care HPV system is feasible for use on a wide scale [67]. Their system consisted of a self-collected point-of-care HPV testing method and same-day treatment strategy, which was found to be effective and safe [67]. This new research is very promising for use in developing countries, however more point-of-care HPV testing platforms must be approved for use in order to be considered for use on a large scale for screening purposes [66,68]. 

While HPV-only testing is the mainstay of new guidelines, cervical cytology is still commonly used in many areas [17,48,69]. Similarly, cotesting with cervical cytology and HPV testing is still performed, which may be indicative of the historical influence that the traditional Pap smear has had due to its longstanding position as the quintessential cervical cancer screening method [56,60,69]. This delay in shift to HPV testing seen may be due to a lag of HPV testing infrastructure [65]. It takes significant time and funding to set up the support required for a new screening method on a national or even regional scale, particularly when one functioning system such as that supporting Pap smear screening is already in place [69]. Cervical cytology has its own set of costly supportive requirements, including sample collection materials, specimen storage medium for liquid-based cytology, laboratories with adequate staining and microscope equipment, and, most critically, trained cytotechnologists and cytopathologists [70]. Screening guidelines do occasionally recommend cotesting with both cervical cytology and HPV testing, such as those based in the United States [53]. Established support for cervical cytology as a screening method may be one explanation as to why the transition to HPV-only testing has been slow on the uptake [69]. However, the well-established evidence to support HPV testing as a sufficient sole primary method of screening for cervical cancer may render cotesting obsolete with time. 

Less-developed countries often have limited access to the resources required to carry out HPV testing or cervical cytology on a large scale [29,65]. Where these modalities are not available, often times VIA becomes the test of choice for cervical cancer screening [71]. This method is advantageous in that it does not require a separate laboratory to process specimens and does not rely on lab results to drive the decision to pursue further workup [72]. A patient may be seen in a provider’s clinic and undergo immediate colposcopy or further intervention based on VIA results in real time [72]. This can be particularly useful when a patient population must travel great lengths to be evaluated for each visit or has low adherence to follow up. Again, a disadvantage to VIA screening is low specificity and potential for redundancy or overtreatment, but this method of screening is still commonly utilized in resource-limited settings [27,71]. 

### 2.3. Screening Parameters: A Barrier to Implementation

With access to resources identified as a limiting factor, it then becomes paramount to employ cervical cancer screening in the right population. Cervical cancer screening is appropriate for any individual with a cervix. Identifying the appropriate age to initiate screening is important to appropriately utilize resources. In some current guidelines, cervical cancer screening is recommended to start as early as 21 years despite the average age of cervical cancer diagnosis being 53 years of age globally (see Table 1) [40,53]. The most common ages to test positive for HPV are bimodally distributed, with the first peak in the 20s, where the infection is more likely to be cleared by the immune system [12]. For these reasons, many screening guidelines have shifted the recommended age to start screening to either 25 or 30 years of age. On the other end of the spectrum, most screening guidelines recommend cessation of screening by age 65 at the latest (see Table 1). The most recent WHO recommendation recommends discontinuation of screening after age 50, provided the individual tests negative for two consecutive tests at 5 and 10-year intervals [48]. 

The appropriate interval between screening also differs between guidelines. Studies have found effectiveness with a longer interval of screening with HPV-testing alone versus cytology alone, based on the lower likelihood of developing cervical atypia or cancer in the setting of a negative HPV test as compared to a negative cervical cytology test [30]. Vaccination efforts affect prevalence of the most carcinogenic HPV strains and contribute to longer screening intervals. Data suggests that a screening interval of every 5–10 years is appropriate in those who test negative for HPV, which decreases burden on both the patient and the health care system while still providing adequate protection for the population [73,74]. If screened with cervical cytology alone, dated data suggests that a screening interval of every 3 years is appropriate in patients who have a history of negative screening tests [75].

### 2.4. Sampling: A Barrier to Implementation

Method of sampling may also become a useful tool in reaching a larger proportion of patients. Traditionally, provider-collected sampling has been the sole method of obtaining specimens to screen patients. Advantages to this method of screening include the remainder of an individual’s gynecologic evaluation—symptomology, breast exam, visualization of anatomy, bimanual exam, and addressing other issues at the visit by the healthcare provider. Some barriers of provider-collected sampling are cost of travel to and establishing care at an equipped healthcare facility, which are frequently large enough hurdles to prevent screening from taking place at a regular interval or at all [76,77]. Other patient factors (personal trauma history, language barriers, etc.) can also significantly decrease an individual’s likelihood to interact with the healthcare system or can negatively affect the quality of the interaction [78,79].

Recently developed self-sampling kits present a unique opportunity for alternative cervical cancer screening in those who do not regularly present for evaluation and screening. Recent studies have shown that the accuracy of self-collected samples has moderate to substantial agreement with provider-collected sampling; self-collected sampling has sensitivities and specificities close to 0.90 [80]. There are several studies investigating the utility of self-sampling in various countries in sub-Saharan Africa and Latin America, as well as in China with initial results being promising for use in low-resource settings [41,42,65,81]. Potential benefits of self-collected sampling include: increased screening access with mail-in infrastructure, decreased need for in-facility visit with trained provider, and decreased patient discomfort with overall good acceptance [82]. Data on self-collected sampling also shows that offering this option to patients increases participation in cervical cancer screening programs, recruiting more patients from those who may not otherwise have undergone screening [83,84]. Patients undergoing self-collected screening would have to be advised to schedule in-person visits for symptoms or problems as this method would not allow for provider evaluation if present. Additionally, if screening tests were to result as positive with further evaluation indicated, the patient would need to schedule an in-person visit for further testing which may be difficult if a relationship has not been established with a provider; ensuring proper follow up in the event of positive testing would be paramount to the function of this screening method [82]. In the future, we envision ‘built in’ molecular triage that will provide extended genotyping or methylation and may further improve the utility of self-sampling.

### 2.5. Provider Buy-In: A Barrier to Implementation

Whether undergoing provider-collected or self-collected sampling, significant forethought must be put into provider education and implementation into clinical practice. Support is needed on a systems level as well: a 2011 Cochrane systematic review noted mailed invitations as effective for increasing absolute uptake of screening [85]. Healthcare institutions, practices, and individuals themselves must be diligent in ensuring continuing medical education (CME) for the most up to date and accurate screening information available. Healthcare systems in the United States frequently build in support for providers to pursue CME; disseminating the updates for cervical cancer screening through CME provides an opportunity for staying on top of new guidelines. The need for this type of education for both providers and patients is exemplified by data showing a significant increase in women in the U.S. with overdue cervical cancer screening between 2005 and 2019, from 14.4% to 23.0% [86]. Provider buy-in of guidelines will be an important factor to consider when creating updated guidelines and working towards global implementation. 

## 3. Recommendations

Cervical cancer remains a significant worldwide health concern. Elimination of this disease is possible with vaccination. Until vaccination uptake is significant, screening by a highly sensitive methodology should be utilized. Screening regimens have become complicated and difficult for providers and patients to understand. The WHO strategy to eliminate cervical cancer (a cervical cancer incidence rate of less than or equal to 4 per 100,000 women) by 2030 is an attainable goal [87].

Recommendations:
-Vaccination of at least 90% of girls by age 15, in alignment with the WHO guidelines.
-In addition to offering vaccination to all individuals ages 9–45.-Universal recommendation of HPV only as the preferred screening modalities.
-Initiate screening between 25–35.-Self-collection as an option for all patients.

Dual-stain testing for those with HPV + disease to triage intervention.

## 4. Conclusions

In conclusion, cervical cancer, while a treatable and preventable disease, continues to impact the health of too many women worldwide. This manuscript summarizes the experiences of countries with various levels of screening and treatment resources. We stand behind the WHO’s global strategy to eliminate cervical cancer by 2030. This will require a multi-pronged approach: Vaccination, Screening and Treatment. The benefits of meeting these goals are lives lived (aversion of over 14 million cervical cancer deaths by 2070) and avoidance of the widespread suffering caused by this disease. 

## Figures and Tables

**Table 1 healthcare-11-02273-t001:** Recommendations for cervical cancer screening by organization.

Organization	Location	Release Date	Recommended Screening Measures	Age of Screening	Additional Considerations
WHO [48]	International	July 2021	Screen with HR-HPV DNA testing every 5–10 years. If HR-HPV DNA testing is not available, the preferred alternative is screening every 3 years with cytology or VIA.	Age 30 to 50	For individuals who are HIV positive, screen every 3 to 5 years with HR-HPV DNA testing preferred starting at age 25. For all individuals, may stop screening after two consecutive negative tests of the appropriate intervals after the age of 50.
FIGO [51,52]	International	May 2021	Approach screening guidelines on a national or regional basis based on resources available.		Recommend that at a minimum, HPV vaccination be provided to young female individuals in the population ages 11 to 18 and followed with at least a single HR-HPV screen at age 35 or 39 years, or at least a single VIA. If resources are available, repeat HR-HPV screening at 10-year intervals or VIA at 5-year intervals.
European Commission [50]	European Union nations	November 2022	Screen with HR-HPV testing every 5 years or more.	Age 30 to 65	Consider offering HPV cervical self-collection kits to those who decline routine evaluation. Consider altering screening age and interval in those who have completed HPV vaccination.
USPSTF [53]	United States	August 2018	In women aged 21 to 29, screen every 3 years with cytology alone. In women aged 30 to 65 years, screen every 3 years with cytology alone, every 5 years with HR-HPV testing alone, or every 5 years with co-testing (both HR-HPV and cytology).	Age 21 to 65	Currently under review as of March 2022 [54].
ACOG [55]	United States	April 2021	Endorses USPSTF recommendations.	Age 21 to 65	
ACS [56]	United States	July 2020	Screen with HR-HPV testing every 5 years. If not available, screen with co-testing with HR-HPV and cytology every 5 years, or with cytology every 3 years.	Age 25 to 65	
ASCO [57]	United States	September 2022	Screen with HR-HPV DNA testing in all resource settings where available. VIA may be used in basic resource settings. Depending on setting ^a^: Screen every 5 years from age 25 to 65 in maximal resource setting. Screen every 5 years from age 30 to 65 in enhanced resource settings, if two consecutive negative tests can extend to every 10 years. Screen every 10 years from age 30 to 49 in limited resource settings. Screen 1 to 3 times per lifetime from ages 30 to 49 years in basic resource settings.	Varies based on resource setting.	In HIV positive individuals, recommend HR-HPV screening immediately after HIV diagnosis, followed by twice as many screenings in a lifetime compared to the general population. Screening is recommended at 6 months postpartum in all resource settings or at 6 weeks postpartum in basic resource settings.
ASCCP [58]	United States	July 2021	Endorses the 2018 USPSTF recommendations and the 2020 ACS recommendations.		Endorses any cervical cancer screening for secondary prevention of cervical cancer and recommends interventions that improve screening for those who are under screened or unscreened.

Abbreviations: American College of Obstetricians and Gynecologists (ACOG); American Cancer Society (ACS); American Society for Colposcopy and Cervical Pathology (ASCCP); American Society for Clinical Oncology (ASCO); The International Federation of Gynecology and Obstetrics (FIGO); Human Immunodeficiency Virus (HIV); high-risk strains (HR); human papillomavirus (HPV); United States Preventive Services Task Force (USPSTF); visual inspection with acetic acid (VIA); World Health Organization (WHO). ^a^ Four-tiered resource setting definitions by ASCO include [57]:^.^ Maximal: High-level/state-of-the art resources or services available; screening guidelines do not adapt to resource constraints. Enhanced: Enhanced-level resources with available services that are optional; screening guidelines allow for individual choice due to increased number and quality of screening and treatment options. Limited: Limited-level services available, limited by financial means and modest infrastructure; guidelines typically feasible for a greater percentage of the population than the primary target group. Basic: Core resources or fundamental services absolutely necessary for any public health/primary health care system to function are available; guidelines typically target the highest-need population.

## Data Availability

No new data were created or analyzed in this study. Data sharing is not applicable to this article.

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
