# Peer review of "Cervical Cancer Screening Recommendations: Now and for the Future"

_healthcare, 2023, doi:10.3390/healthcare11162273_

Round 1

Reviewer 1 Report

Thank you for the opportunity to review this manuscript. The goal of screening for cervical cancer is to find precancerous cervical cell changes when treatment can prevent cervical cancer.  Recently WHO has updated its recommendations for the human papillomavirus (HPV) vaccine. However, I'm confused about the purpose of this study. It's necessary to better to specify the aim of this study in the introduction. Is it a summary of recent evidence about cervical cancer screening, right? Why was it needed? I think the discussion could present the necessity to integrate health services to offer the possibility of performing cervical screening in other contexts. Here some interesting reference "Sinopoli A, Baccolini V, Di Rosa E. Killing Two Birds with One Stone: Is the COVID-19 Vaccination Campaign an Opportunity to Improve Adherence to Cancer Screening Programmes? The Challenge of a Pilot Project in a Large Local Health Authority in Rome. Vaccines. 2023; 11(3):523. https://doi.org/10.3390/vaccines11030523", "Staley H, Shiraz A, Shreeve N, Bryant A, Martin-Hirsch PP, Gajjar K. Interventions targeted at women to encourage the uptake of cervical screening. Cochrane Database Syst Rev. 2021 Sep 6;9(9):CD002834. doi: 10.1002/14651858.CD002834.pub3. PMID: 34694000; PMCID: PMC8543674."

Minor editing of English language required

Author Response

Response: Thank you for your comments. This is a review of the current state of evidence rather than a traditional research study. The purpose of the study is clarified here:

Lines 150-153: “Today, guidelines for the prevention and screening for cervical dysplasia and malignancy have become convoluted and difficult for practitioners to comprehend. This article aims to discuss the current state of cervical cancer screening guidelines and predict the changes that are likely in the future.”

Reviewer 2 Report

This review is well-written and provides a comprehensive overview of the cervical screening situation.  I only have 1 main concern.  The authors have put a lot of emphasis on dual stain for triaging high-risk HPV women.  Most countries , especially in Europe, are not recommending dual stain as a standard method for triaging HR HPV women yet.  I would suggest the authors discuss further other triaging methods eg HPV 16/18 genotyping, DNA methylation as well. It may be not be appropriate to include dual staining as part of the final recommendations in the manuscript.

It may be interesting to include a slightly more detailed discussion on the pros and cons regarding the use of HPV alone vs co-testing ( HPV + cytology ) as the primary screening method.

The authors may also consider adding some discussion on the use of valid HPV tests as this is also an issue globally, where many non-validated HPV tests are emerging in the market. 

One minor point - it may be better to refer to the screening population as " women/ individual" instead of as " patients" eg in line 299 etc. These women are not ill, so may not be appropriate to call them " patients".

Author Response

Reviewer #2, Comment 2:

One minor point - it may be better to refer to the screening population as " women/ individual" instead of as " patients" eg in line 299 etc. These women are not ill, so may not be appropriate to call them "patients".

Response:

The term “patient” is used in this portion of the article to identify individuals interacting with the healthcare system for their own medical care while “provider” is a term used to identify individuals administering medical care through ordering tests, triaging, etc. We believe it is important to make this distinction and use dedicated terms rather than frequently using “individual” which may be interpreted in a less specific manner.

Reviewer 3 Report

Excellent review article. It covers all relevant aspects of the cervical cancer prevention problems. In our studies on the Polish women population we observe a different frequency of HPV types. Type 18 seems to be less important than assumed. Therefore, in the future, it is worth analyzing the composition of HPV vaccines.

Author Response

Reviewer #3, Comment 1:

Excellent review article. It covers all relevant aspects of the cervical cancer prevention problems. In our studies on the Polish women population we observe a different frequency of HPV types. Type 18 seems to be less important than assumed. Therefore, in the future, it is worth analyzing the composition of HPV vaccines.

Response:

Thank you for the review and the point regarding HPV type prevalence. We agree, this is something interesting to follow in the future.

Round 2

Reviewer 1 Report

This version is ok for me.